# Zoning for Spatial Conservation and Restoration Based on Ecosystem Services in Highly Urbanized Region: A Case Study in Beijing-Tianjin-Hebei, China

**Wen Zhou** [1,2], **Yantao Xi** [2], **Liang Zhai** [1], **Cheng Li** [2], **Jingyang Li** [3] **and Wei Hou** [1,*]

1 Chinese Academy of Surveying and Mapping, Lianhuachi West Road 28, Beijing 100830, China
2 School of Resources and Geosciences, China University of Mining and Technology, Xuzhou 221116, China
3 Department of Housing and Urban-Rural Development of Shanxi Province, Taiyuan 030013, China
* Correspondence: houwei@casm.ac.cn

**Abstract:** Ecosystem services are highly affected by human activities, especially in the fast-urbanizing regions. It is important that the regional development or urbanization strategy be implemented by maintaining or protecting the long-term provision ability of multiple ecosystem services. The spatial pattern of ecosystem services and zoning for spatial conservation and restoration are the preconditions of sustainable development. With the Beijing-Tianjin-Hebei (BTH) region as the research area, an approach for spatial zoning was proposed on the basis of the modeling results of key ecosystem services (water retention, soil retention, heat mitigation, and carbon storage). Our results show that the hot spots of ecosystem services are mainly in the north and the west at high altitudes and with large vegetation coverage, while the cold spots are mainly in the plain area of the southeast in the BTH region. In addition, the whole region is divided into five ecological zones: the ecological restoration zone, ecological transition zone, coastal ecological protection zone, soil and water retention zone, and ecological security shelter. Each zone has applied different strategies for ecological restoration and conservation. The results represent the spatial heterogeneity and major functions in different zones, and they can provide planning guidance for supporting the coordinated development of the BTH region.

**Keywords:** ecosystem services; hot spot analysis; ecological zoning; Beijing-Tianjin-Hebei region



## 1. Introduction

Rapid urbanization makes a significant contribution to social improvement and economic development while posing a great threat to the health of natural ecosystems [1]. With the expansion of cities and towns, the ecological space has continuously been shrunk, fragmented, and isolated. The structure and the function of ecosystems have also undergone tremendous changes [2,3]. The concept of ecosystem services was proposed for understanding how ecosystem processes directly sustain or benefit human life [4–6]. According to the 2019 Global Assessment of Biodiversity and Ecosystem Services, 75% of the global terrestrial ecosystems had been significantly altered by human activities to date, resulting in a rapid decline in most ecosystem services and biodiversity [7]. The degradation of ecosystem services will in turn affect human well-being and socioeconomic development [8]. In order to implement differentiated policies of ecosystem conservation and restoration in their proper places, ecological zoning based on precise simulations of ecosystem services is needed to clarify the regional functions and spatial heterogeneity [9,10]. With additional consideration on social and economic conditions, strategies and suggestions can be proposed to efficiently guide spatial planning and promote sustainable development in the region [11–13].

Many scholars have conducted research related to ecosystem service assessment and spatial zoning. Methods of quantitative analysis, e.g., statistical methods [14,15], empir-

ical approaches [16,17], and ecological models [18–20], are adopted for evaluating the spatial and temporal characteristics of ecosystem services. In terms of spatial zoning, the frequently used methods include principal component analysis [21], clustering analysis [12,22], and self-organizing neural network analysis [23,24]. Previous studies have made significant progress in the theory, method, and practice of ecosystem service simulation and assessment and have featured in-depth discussions on the relationship between ecosystem services [8,25–27]. However, most studies have been limited by the relatively low resolution and accuracy of geospatial data, especially in study areas in China. As the spatial distribution of ecosystem services are highly related to land covers [28,29], much more detailed and fine-resolution land-cover data can help to reflect the impact of subtle differences in land use on ecosystem service assessment. Therefore, it would be more suitable for function-oriented zoning, which has been considered as an important planning strategy in China.

In 2014, the coordinated development of the Beijing-Tianjin-Hebei (BTH) region has been announced as a national development strategy of China. The subsequently released "Outline of the Plan for Coordinated Development for the BTH Region" pointed out that "without the integration of ecological and environmental protection, there would be no integration of the BTH region" [30]. In 2019, the state council of the central government recommended strengthening ecological environment protection and management through spatial zoning and by optimizing the structure and layout of the territory for establishing an integrated spatial planning system [31]. After a decade of rapid urbanization, the BTH region is now facing serious environmental problems, such as water shortages, soil erosion, and increasing carbon emissions year on year. Such issues have caused the level of ecological conservation in the BTH region to lag behind the requirements of sustainable development [32–34]. In this research, the administrative unit of county will be used for spatial zoning since it has been regarded as an important carrier for implementing strategic planning [35]. We adopt the latest and most-detailed land-cover data from the national project Geographical Conditions Monitoring (GCM), which is more suitable for spatially modeling ecosystem services [36]. Next, we used the InVEST (integrated valuation of ecosystem services and tradeoffs) models [37] to spatially quantify and map four selected ecosystem services (including water retention (WR), soil retention (SR), heat mitigation (HM), and carbon storage (CS)) that are closely related to ecological conservation and restoration in the BTH region. On this basis, function-oriented zoning is applied through the aggregation of the hot spots/cold spots of various ecosystem services at the county level. The results could better reflect the heterogeneity of the ecological pattern of the BTH region. Our aim is to develop a spatial-zoning approach that is based on the major functions provided in each zone and then to lay a solid foundation for guiding strategic development and optimizing the spatial pattern for the coordinated development of the BTH region.

## 2. Materials and Methods

### 2.1. Study Area and Data Sources

The BTH region is in the heart of the Bohai Sea in northern China, including two municipalities, Beijing and Tianjin, and 11 other prefecture-level cities in the province of Hebei (Figure 1). It has a temperate monsoon climate and covers about 216,900 km$^2$, accounting for 2.3% of the total land area of China [38]. In terms of topography, the altitude gradually decreases from the northwest to the southeast in the BTH region, with the Bashang Plateau in the north, the Yanshan-Taihang Mountains in the northwest, and the coastal plain in the southeast. Important reservoirs such as Miyun and Huairou are in the northeast of the BTH region. In 2018, the total GDP of the BTH region reached CNY 8.5 trillion, accounting for 9% of the national GDP [39–41]. Since the promotion of the national strategy for the coordinated development of BTH, the regional linkages of the three places have been strengthened, and the integrated spatial planning of ecological protection has been emphasized. How to protect and restore the ecological environment and how to

promote the harmonious coexistence of economy development and ecosystem protection are the main challenges.

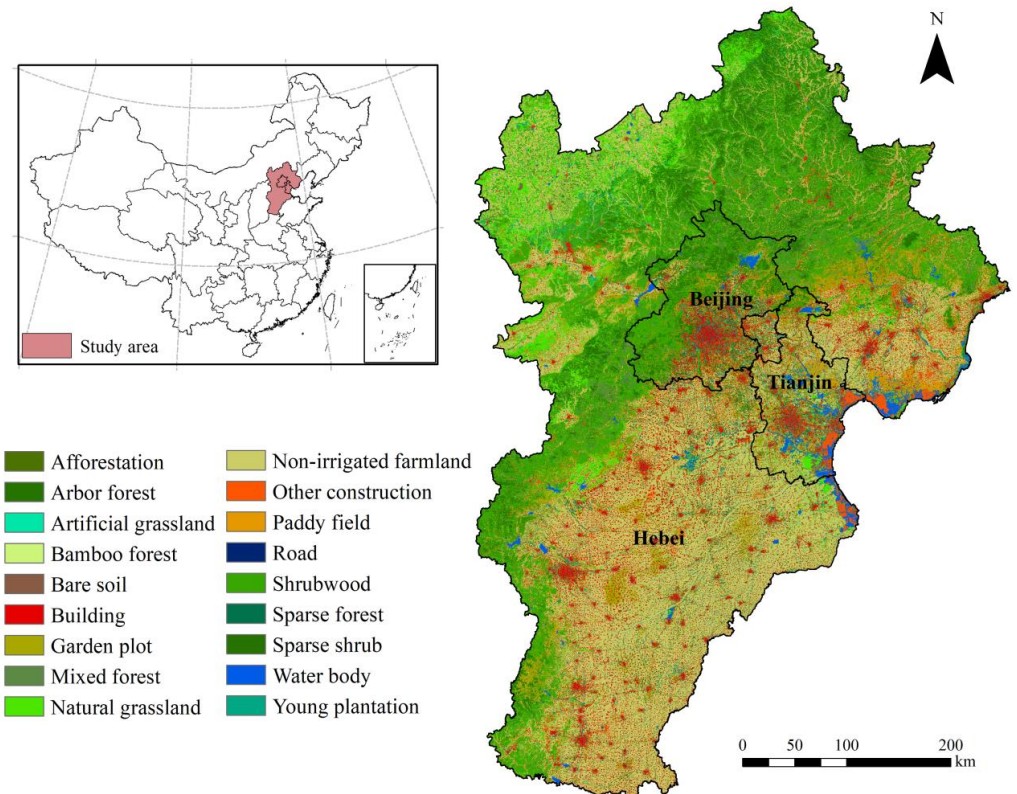

**Figure 1.** Geographical location and land-cover types of BTH.

The main data sources used in this study include the detailed land-cover map, the digital elevation model (DEM), meteorological data, soil data, and socioeconomic data on the BTH region in 2018. The land-cover map of GCM (including 18 land-cover classes) was interpreted by using a very high-resolution remote-sensing image (better than 1 m), which allowed us to evaluate ecosystem service at a very detailed level (Figure 1). DEM data were downloaded from the United States Geological Survey (USGS, https://earthexplorer.usgs.gov/ (accessed on 4 March 2023)), with a resolution of 30 m (Figure 2a). Precipitation data were from the China Meteorological Data Service Centre (CMDC, http://data.cma.cn/ (accessed on 4 March 2023)) with a resolution of 1 km (Figure 2b). The potential evapotranspiration was derived from the National Earth System Science Data Center (http://www.geodata.cn/ (accessed on 4 March 2023)), with a resolution of 1 km (Figure 2c). The soil map (with a resolution of 1 km) and its related attribute data came from the Harmonized World Soil Database (HWSD, https://webarchive.iiasa.ac.at/Research/LUC/External-World-soil-database/HTML/HWSD_Data.html (accessed on 4 March 2023); see an example in Figure 2d). Other socioeconomic data, such as population size and GDP, were obtained from the local Statistical Yearbook [39–41]. All spatial data were resampled to a resolution of 30 m.

### 2.2. Ecosystem Services Modeling

The key ecosystem services are selected on the basis of the natural conditions and the critical environmental issues in the BTH region. For example, the Yanshan-Taihang Mountains, which are the source of the Haihe River, are important areas for water retention, and the forest ecosystem in the mountainous area of north and northwest plays a crucial role in carbon storage. The soil and rock surfaces and loess hills in this region are the key control areas of soil and water loss [42]. The expansion of human construction has led to

forest and wetland degradation and fragmentation [43,44], as well as water shortages in the Miyun Reservoir and the Chaohe River Basin [45]. In general, intensified human activities in BTH have exerted significant pressure on the environment and related risks of climate change [46]. In view of this, four ecosystem services (i.e., water retention, soil retention, heat mitigation, and carbon storage) were selected on the basis of the urgent needs of ecosystem services and data availability and then spatially quantified by InVEST, a widely used tool. This tool provides effective models for the quantitative analysis of ecosystem services. The models and detailed parameters are provided in the Supplementary Document.

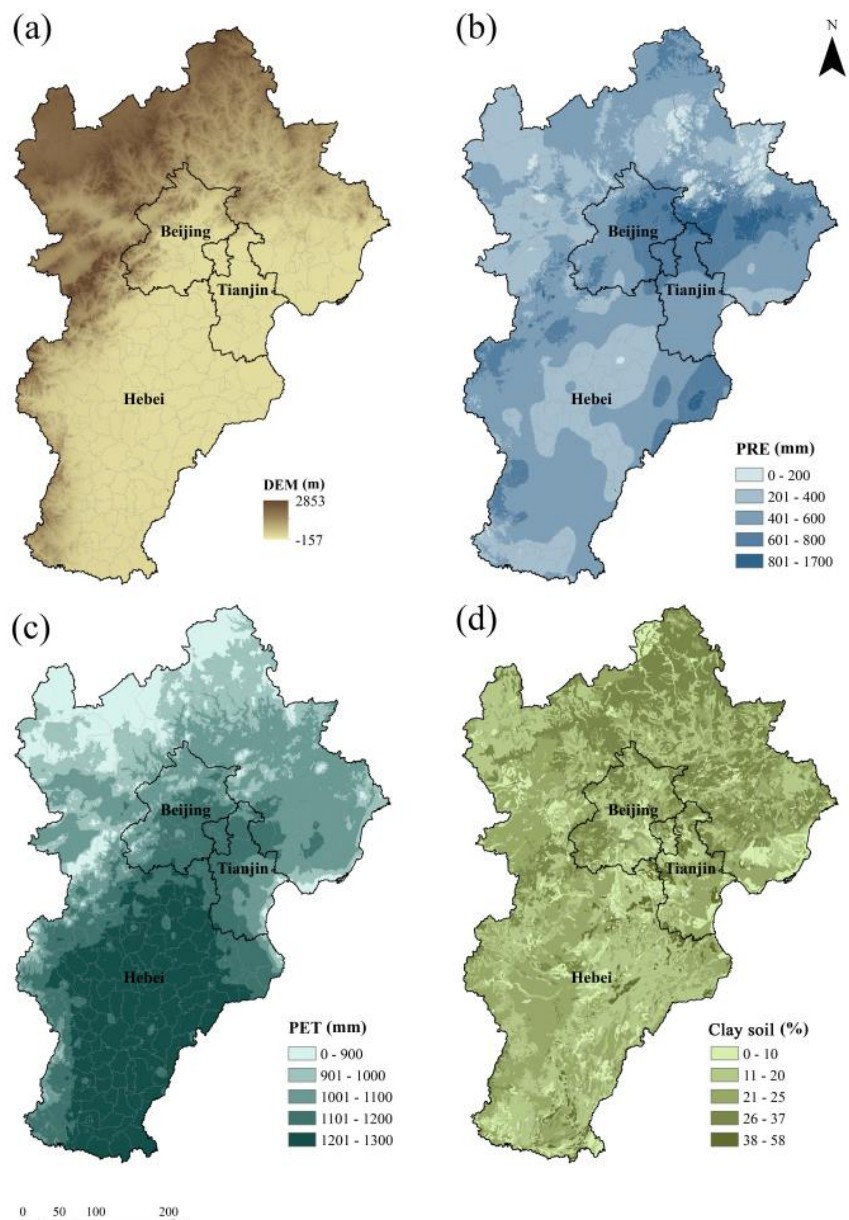

**Figure 2.** Data examples used in the study: (**a**) digital elevation model (DEM), (**b**) precipitation (PRE), (**c**) potential evapotranspiration (PET), (**d**) clay soil percentage.

### 2.3. Hot Spot Analysis

Hot spot analysis was frequently used to identify priority protection areas of ecosystem services [47]. The $G_i^*$ coefficient proposed by Getis and Ord is used to explore the location of hot spots [48]. ArcGIS tool zonal statistics were used to calculate the average value of the four services of each county in the BTH region. Next, $G_i^*$ statistics were adopted to identify

the spatial aggregation areas that can provide large amounts of ecosystem services. The hot spot $G_i^*$ calculation formula is as follows:

$$G_i^* = \frac{\sum\limits_{j}^{n} w_{ij} x_j}{\sum\limits_{j}^{n} x_j} \tag{1}$$

The statistical significance of $G_i^*$ is tested by standardized Z values:

$$Z(G_i^*) = \frac{\sum\limits_{j}^{n} w_{ij} x_j - \overline{x} \sum\limits_{j=1}^{n} w_{ij}}{S \sqrt{\frac{[n\sum\limits_{j}^{n} w_{ij}^2 - (\sum\limits_{j=1}^{n} w_{ij})]^2}{n-1}}} \tag{2}$$

where $w_{ij}$ is the spatial weight between county i and j; $x_j$ is the ecosystem service value of county j; $\overline{x}$ is the average value of ecosystem services in the BTH region; $S$ is the standard deviation of the ecosystem services of the BTH region; and $n$ is the total number of counties. The $G_i^*$ statistic distinguishes between hot spots and cold spots with the varying significance of clustering [49]: when $Z > 2.58$, it is a hot spot; when $1.65 < Z \leq 2.58$, it is a sub-hot spot; when $-1.65 \leq Z \leq 1.65$, it is not significant; when $-2.58 < Z \leq -1.56$, it is a sub-cold spot; when $Z \leq -2.58$, it is a cold spot. The hot spots represent aggregated pixels with higher values of ecosystem services compared with the surrounding area. On the contrary, the cold spots represent aggregated pixels with lower values.

### 2.4. Correlation Analysis and Functional Zoning

Spatial correlation is usually used to evaluate and describe whether there is spatial correlation and a degree of correlation between spatial objects [50]. Moran's I index is a kind of spatial autocorrelation coefficient, and it can be used to explore the spatial relationship between the four ecosystem services in the BTH region. The Moran's I index ranges from $-1$ to 1. If the index value is greater than 0, the two types of service are positively correlated; if the index value is less than 0, the two types of service are negatively correlated. Additionally, the correlation significance can be quantified by using the *p*-value, which stands for the probability [51]. If the *p*-value is very small, it means that it is very unlikely (small probability) that the observed spatial pattern is the result of random processes. For example, when the *p*-value equals to 0.01, this indicates that the elements are a concentrated distribution at a 99% confidence level [52].

Cluster analysis is a comprehensive analysis of the characteristics of the data. K-means clustering is an unsupervised classification method for achieving the maximum dissimilarity among the clusters and has been widely used in the study of geographical spatial patterns [53]. In this study, we adopted Geoda software [54] for spatial zonation at the county level in the BTH region. The K-means algorithm supplied by Geoda was applied to the results of the hot spot analysis for various ecosystem services, to identify the zones with the distinct spatial patterns of the four ecosystem services through the iteration process. The number of clusters was decided by considering the average variances of ecosystem services in each zone as a minimum.

## 3. Results

### 3.1. Spatial Pattern of Ecosystem Services

In 2018, four ecosystem services, namely WR, SR, HM, and CS, were calculated at the pixel level and then aggregated for each county in the BTH region (Figure 3). The overall provisions of the ecosystem services in the BTH region were $97.4 \times 10^8$ m$^3$ (WR), $5.53 \times 10^8$ t (SR), and 2673 tg (CS), respectively, and the mean value of the HM index (HMI) was 0.57 (Table 1). The high-value areas of WR were distributed mainly in the counties



in the northern region. The spatial distribution patterns of SR, HM, and CS were similar, where the high-value areas were mainly in the northern and western mountainous areas with large vegetation coverage and fewer human disturbances, and the low-value areas were mainly in the plain areas with intensive human activities and high urbanization levels.

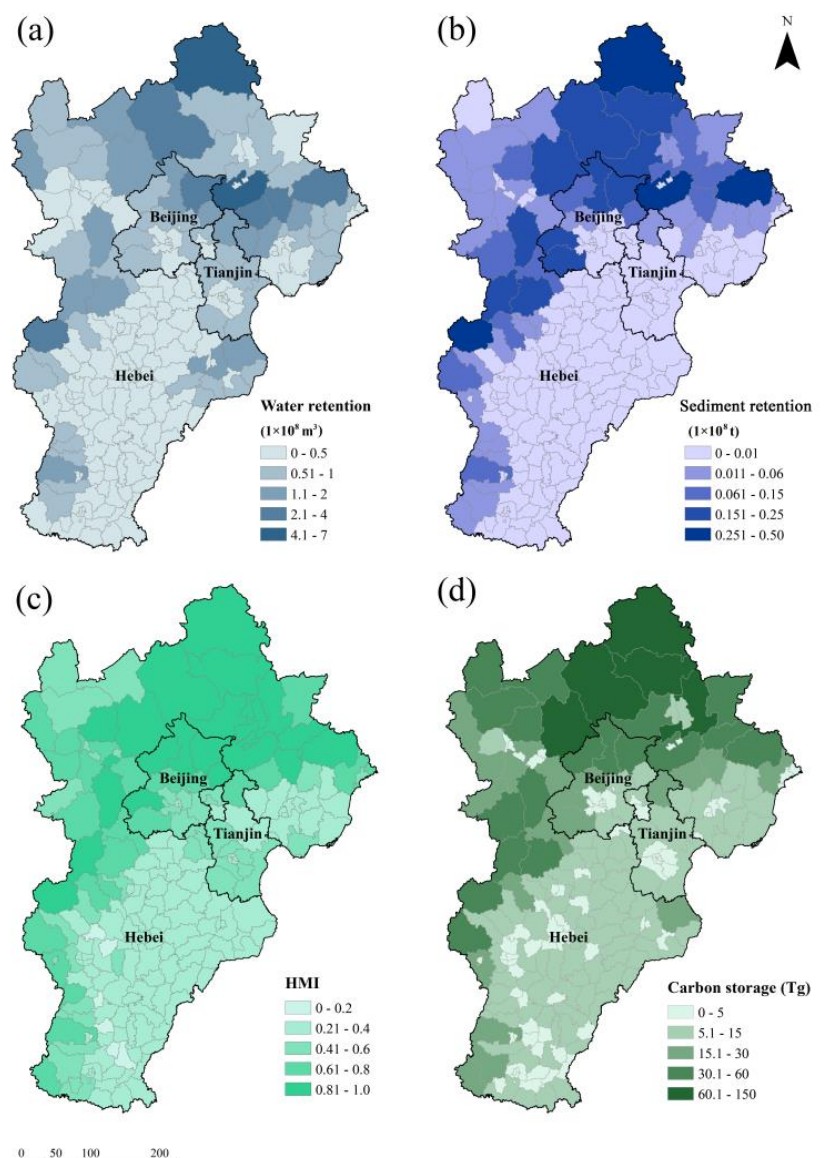

**Figure 3.** Spatial distribution of ecosystem services at the county level in the BTH region, including water retention (**a**), sediment retention (**b**), heat mitigation (**c**), carbon storage (**d**).

**Table 1.** Statistics of the modeling results of ecosystem services in the BTH region.

| Region | WR | | SR | | HMI | | CS | |
| --- | --- | --- | --- | --- | --- | --- | --- | --- |
| | Value ($1 \times 10^8$ m³) | Ratio [1] (%) | Value ($1 \times 10^8$ t) | Ratio (%) | Mean Value | Ratio (%) | Value ($1 \times 10^2$ tg) | Ratio (%) |
| Beijing | 10.90 | 11.18 | 1.07 | 19.32 | 0.77 | 135.88 | 2.36 | 8.84 |
| Tianjin | 6.50 | 6.66 | 0.06 | 1.17 | 0.45 | 79.41 | 1.00 | 3.73 |
| Hebei | 80.00 | 82.16 | 4.40 | 79.51 | 0.48 | 84.71 | 23.37 | 87.43 |
| BTH | 97.40 | 100 | 5.53 | 100 | 0.57 | - | 26.73 | 100 |

[1] Ratio represents the ecosystem service value of a region to total value (or mean value for HMI) of BTH.

### *3.2. Hot Spots of Ecosystem Services*

The spatial patterns of the hot spots of different ecosystem services were distinct in the BTH region (Figure 4). The area of the WR hot spots was 42,011.68 km², accounting for 19.46% of the study area (Table 2). They were mainly in the central and western regions, covering the counties of western Beijing, northern Tianjin, and western Hebei. The spatial locations of the SR hot spots were situated in the northern mountainous areas. The total area of the SR hot spots accounted for 23.22% of the BTH region. Additionally, the hot spots of CS and HM accounted for 29.54% and 38.15% of the region, respectively. Their spatial patterns were similar, and hot spots were mainly in the north of Hebei. In general, the spatial distributions of the four ecosystem services were mainly in the north and west of the BTH region at high altitudes and with large vegetation coverage, while the cold spots were mainly in the plain areas of the southeast of the BTH region.

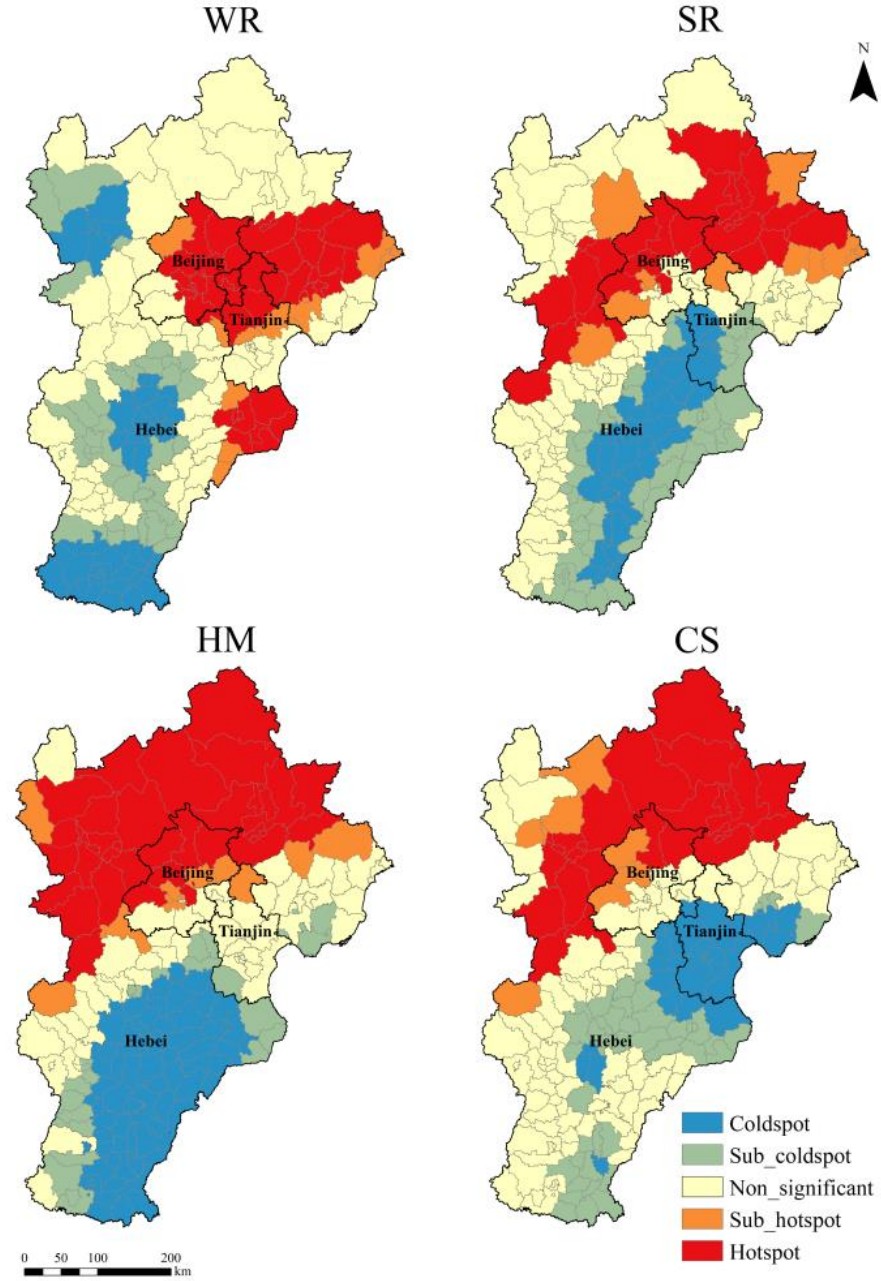

**Figure 4.** Spatial hot spots and cold spots of ecosystem services in the BTH region, including water retention (WR), soil retention (SR), heat mitigation (HM), and carbon storage (CS).

**Table 2.** Statistics of hot spots and cold spots of ecosystem services.

| Type | Hot Spots | | Sub-Hot Spots | | Cold Spots | | Sub-Cold Spots | |
| --- | --- | --- | --- | --- | --- | --- | --- | --- |
| | Area (km$^2$) | Proportion (%) | Area (km$^2$) | Proportion (%) | Area (km$^2$) | Proportion (%) | Area (km$^2$) | Proportion (%) |
| WR | 42,011.68 | 19.46 | 10,226.48 | 4.74 | 29,738.66 | 13.77 | 31,743.87 | 14.70 |
| SR | 50,146.63 | 23.22 | 19,479.60 | 9.02 | 27,998.22 | 12.97 | 32,201.35 | 14.91 |
| HM | 82,381.41 | 38.15 | 16,142.63 | 7.48 | 46,118.88 | 21.36 | 18,447.28 | 8.54 |
| CS | 63,793.91 | 29.54 | 14,517.87 | 6.72 | 24,277.37 | 11.24 | 29,021.51 | 13.44 |

*3.3. Spatial Correlation and Zoning*

3.3.1. Spatial Correlation

The Moran's I index calculated for each pair of ecosystem services is shown in Table 3. Our results demonstrated that any two ecosystem services were spatial correlated as all indexes were greater than 0 and pass the significance test ($p < 0.01$), with the confidence of 99%. The Moran's I index values for the SR-HM, SR-CS, and HM-CS were greater than 0.5, which means that their correlations were extremely significant, followed by the correlations of WR-SR and WR-HM. Overall, the correlations between WR and the three ecosystem services were relatively weak compared with the others.

**Table 3.** Moran's I index of ecosystem services in the BTH region.

| Ecosystem Services | Moran's I | *p*-Value |
| --- | --- | --- |
| WR-SR | 0.2451 | 0.0010 |
| WR-HM | 0.1416 | 0.0010 |
| WR-CS | 0.0732 | 0.0080 |
| SR-HM | 0.5581 | 0.0010 |
| SR-CS | 0.5409 | 0.0010 |
| HM-CS | 0.5635 | 0.0010 |

3.3.2. Spatial Zoning in the BTH Region

Only the hot spots of ecosystem services were used in spatial clustering because the cold spots were not significantly related to function-oriented zoning. According to the characteristics of ecosystem services supplied in each cluster, five functional zones were identified, namely the ecological restoration zone (ERZ), ecological transition zone (ETZ), coastal ecological protection zone (CPZ), soil and water retention zone (SWRZ), and ecological security shelter (ESS) (Figure 5). The pie chart in Figure 5 represents the area ratio of hot spots and sub-hot spots of ecosystem services in different zones to the total area of the zone, which can reflect the contribution of different services to the zone.

The ecological restoration zone was mainly in the southeastern plain area of BTH, including 65 counties with a total area of 41,319 km$^2$, accounting for 19.13% of the study area. The provision ability of the four ecosystem services in this zone is very low. Hot spots and sub-hot spots for water retention accounted for 3% of the ecological restoration zone, while the other three services accounted for 0%. The ecological transition zone was mainly in the southwestern mountainous area and the foothills, including 46 counties with a total area of 50,809 km$^2$, accounting for 23.53% of the study area. Of the four ecosystem services, hot spot and sub-hot spot areas for heat mitigation accounted for the most, reaching 27%. The coastal protection zone was in the eastern part of the study area around the Bohai Sea, including 34 counties with a total area of 25,684 km$^2$, accounting for 11.89%. The tradeoff between water retention and the other three services was particularly prominent. Soil and water retention zone was mainly in the foothills of the northern part of the BTH including 31 counties with a total area of 23,458 km$^2$, accounting for 10.86% of the study area. The supply of water retention in this zone was the highest, followed by soil retention, indicating that the area had a strong ability to retain runoff and prevent soil erosion. The ecological

security shelter was mainly in the northwestern mountainous area with high vegetation coverage and low impact from human activities, including 28 counties with a total area of 74,670 km², accounting for 34.58% of the study area. In this zone, the supply of various ecosystem services was high. The hot spots for carbon storage, heat mitigation, and soil retention spatially overlapped, while the hot spots for water retention were lower than the others.

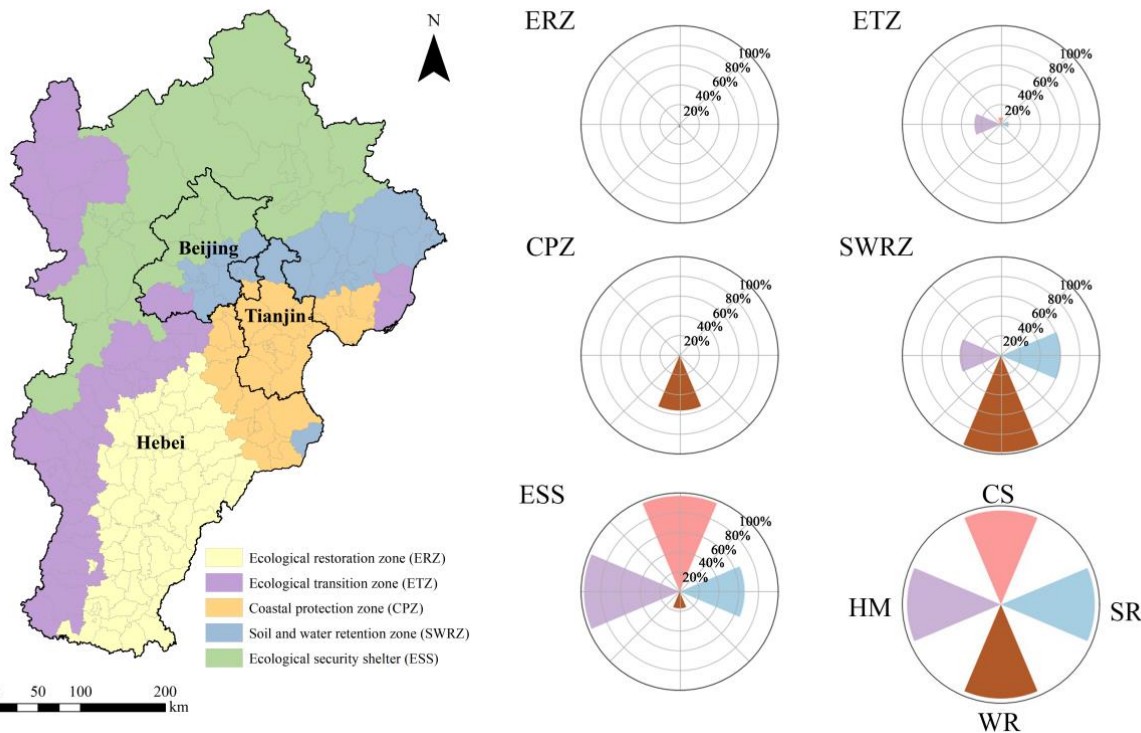

**Figure 5.** Spatial pattern of ecological zones of the BTH region and the proportion of hot spots and sub-hot spots of ecosystem services in each zone.

Figure 6 shows the statistics of ecosystem services in different ecological zones. The value of the WR in the ecological restoration zone was the lowest among all ecological zones. Counties distributed in the Yanshan-Taihang Mountains at high altitudes and with rich vegetation types showed high average water retention values. In terms of the SR and CS, their mean values in the zone of the ecological security shelter were the highest, indicating the high conservation priority in this zone. These services in other regions were significantly lower, which means that certain ecological restoration measures are needed to improve their provision capacity of the services. With regard to the HM, the service provision in the zone of the ecological security shelter was also the highest and relatively high in the ecological transition zone and the soil and water retention zone. On the whole, the average values of ecosystem services in the ecological restoration zone were much lower than those of other ecological zones. Therefore, future developments in the counties in this zone should focus on ecological restoration and proper planning to improve their environments. According to the normalized mean value of ecosystem services in each zone (Figure 6e), the carbon storage values within the five zones were higher than the other ecosystem services, indicating the importance of the CS service to the sustainable development of the region. Except for the zone of ecological security shelter, the SR values were low in other four zones, indicating that soil conservation services in the BTH region were the keys to ecological restoration.

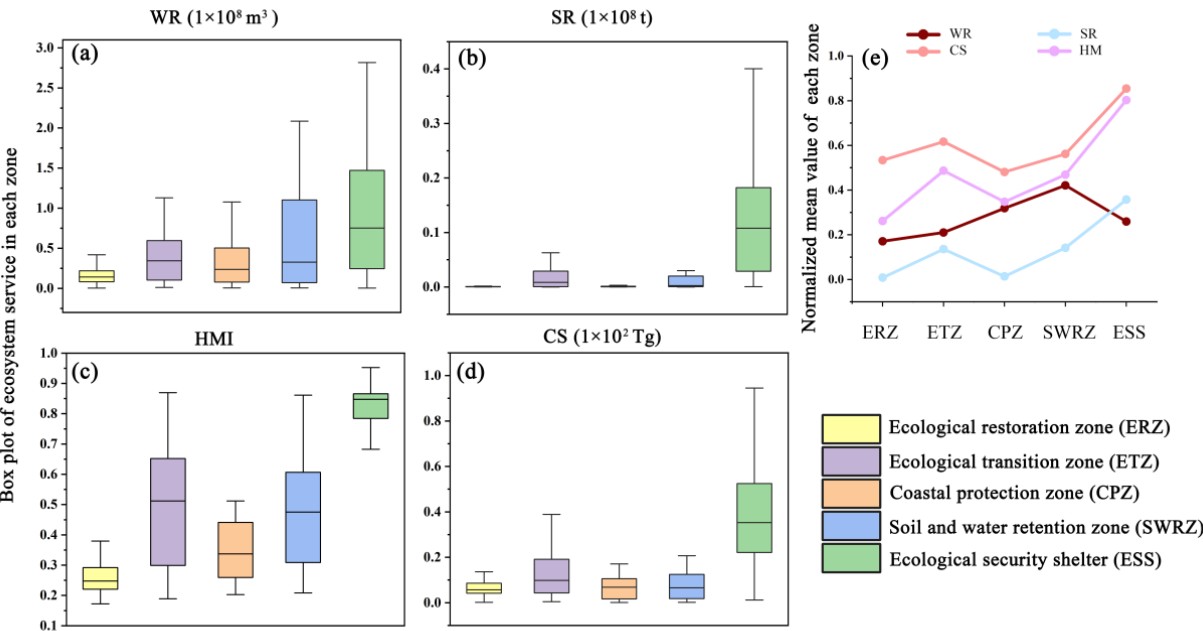

**Figure 6.** Statistics of ecosystem services in different ecological zones (**a**–**d**) and comparison of the normalized mean value of each zone (**e**).

## 4. Discussion

### 4.1. Conservation and Restoration Strategies of Different Ecological Zones

The landscape pattern of the BTH region is diverse, with vegetation covers mostly in the northwest and built-up areas in the southeast. Such a pattern is reflected in the findings of our ecosystem service assessment. To further analyze the natural and social conditions of different ecological zones, we calculated the average values of six factors (precipitation, DEM, temperature, population, proportion of construction land area, and GDP) that can affect the provision ability of ecosystem services [55]. Next, the statistics (see Table 4) were combined with the major functions (the hot spots of ecosystem services) in five zones, where the development strategy for each zone is proposed in a targeted manner to achieve proper conservation and restoration.

**Table 4.** Mean statistics of natural and artificial factors in different zones.

| Ecological Zones | Natural Factors | | | | Artificial Factors | |
|---|---|---|---|---|---|---|
| | Precipitation (m$^3$) | DEM (m) | Temperature (° C) | Population Density (People/km$^2$) | Proportion of Built-Up Area (%) | GDP (CNY 100 Million) |
| ERZ | 413.82 | 36.86 | 27.40 | 1319.79 | 0.21 | 146.97 |
| ETZ | 440.73 | 382.42 | 25.30 | 1187.09 | 0.20 | 209.99 |
| CPZ | 532.66 | 10.59 | 27.52 | 5237.58 | 0.31 | 474.01 |
| SWRZ | 590.44 | 78.92 | 26.83 | 4787.91 | 0.34 | 1325.76 |
| ESS | 439.21 | 811.91 | 22.43 | 275.27 | 0.06 | 170.12 |

The ecological restoration zone was mainly in the plain area at a low altitude, and the precipitation was lower here than in the other four zones. However, the average temperature was relatively high compared with the whole region. There was almost no hot spot of ecosystem services in this zone. Although the proportion of built-up area reached 21%, the GDP was smaller than other zones. Under the premise of satisfying economic development, this zone should promote urban structures by improving the green spaces within built-up areas, such as the construction of more urban parks, country-level nature reserves, and green corridors [33]. The aim is to optimize the layout of urban patterns and

the distribution of industries in order to increase the number of ecosystem services and continuously improve the well-being of human life.

The ecological transition zone was mainly in the western region at a high altitude. Its population density and its proportion of built-up area were similar to those of the ERZ, but the average GDP was higher than that of the ERZ. The average temperature was relatively low in this zone. This may be related to the high altitude. Most ecosystem services (except HM) in this zone were not sufficient. The local authorities should pay more attention to retaining the original natural ecosystem to improve the provision of ecosystem services by implementing a reclamation strategy of returning farmland to forest and grass to form a mixed vegetation structure [56].

The coastal protection zone was located in the eastern coastal area at the lowest average altitude. The population density was highest among the five ecological zones, and the proportion of built-up area was also relatively high (31%). At the same time, the precipitation in 2018 was also higher than that of the ETZ and that of the ERZ. The area proportion of the WR hot spots in this zone was about 60%, and other services were almost 0%, indicating that the main function of this zone was water conservation. The development strategy should focus on restoring and expanding coastal wetlands, improving the ecological environment of estuaries, strengthening the role of coastal shelterbelts, and increasing the naturalness of shorelines. In addition, this zone can become an important carbon pool by promoting wetland restoration projects in sea islands and coastal zones [57], as the coast wetland can absorb large amounts of carbon dioxide from the atmosphere and store it for the long term.

The soil and water retention zone was mainly in northern Beijing and Tianjin and in northeastern Hebei. This zone had the highest yearly precipitation (590.44 m$^3$), a relatively low altitude (78.92 m), and a high population density. The proportion of built-up area and the GDP were the highest among the five zones. In terms of ecosystem service provision, the hot spots of most ecosystem services (e.g., WR, SR, and HM) accounted for a relatively high proportion in this zone. In particular, the water conservation service had an absolute advantage in that there were several important water reservoirs in this zone, such as the Miyun Reservoir in Beijing, the Yuqiao Reservoir in Tianjin, the Douhe Reservoir, the Daheting Reservoir, and the Panjiakou Reservoir in Hebei. At the same time, this zone was also an important economic development area with an adequate water supply [58]. The local governments should continue to take soil and water conservation as the basis to ensure the sustainable provision of the existing ecosystem services while increasing the local green spaces to improve the ability of carbon storage to balance the relationship between economic development and ecological protection.

The ecological security shelter had the best ecological condition in the BTH region. It was distributed mainly in the northern mountains at the highest average altitude and with large vegetation coverage and the lowest average temperature. This zone was less affected by human activities, as its population density and its proportion of built-up area were the lowest. Its natural landscape was better maintained than that of the other zones. The distribution of hot spots of each ecosystem service were relatively balanced. However, the ecological vulnerability of this zone has been increasing because of urbanization process [46]. Therefore, it is important to avoid excessive population growth, urban sprawl, and the implementation of new pollution-generating industries in this zone. An optimized development strategy should be adopted to adjust the industrial structure and strictly control the scale of construction. The reasonable development and protection of natural ecosystems should be emphasized in this zone to ensure high-quality ecosystem services. Ecological restoration should focus on the enhancement of water retention.

### 4.2. Limitations and Future Perspectives

In this study, we focus on the regulation and maintenance services that are generally considered for ecological compensation and conservation. The key ecosystem services were selected based on the urgent needs for the ecosystem services of the BTH region,

as well as the available data. Other important regulation services, i.e., windbreak and sand fixation, can be further integrated for spatial zoning that is based on a cumulative analysis with other ecosystem services. Although we have used the best available data for modeling ecosystem services, the resolution of the spatial data is not unified, and other coarse data (e.g., the soil map) may result in ambiguity when matching the spatial data at a high resolution. Furthermore, this research lacks long-term observation results. The ecosystem service of WR can be affected by the climatic factors of the research period and shows slightly different patterns in other years. In order to ensure sustainable urban planning, more models of ecosystem services should be applied for spatial optimization at a much more detailed grid scale to meet the local demands of various ecosystem services.

## 5. Conclusions

In this study, four ecosystem services (water conservation, soil retention, urban heat mitigation, and carbon storage) were spatially quantified and mapped in the BTH region. The hot spots of each service and the spatial correlation between ecosystem services were analyzed. The results showed that the provision of ecosystem services in the northwestern and northern mountainous areas of the BTH region was better than that in the southern plain area. The hot spots of the ecosystem services were concentrated in the northwest region, which features less human disturbance. On this basis, the BTH region was divided into five ecological zones through spatial clustering. Different development strategies are given for each zone on the basis of their respective spatial patterns of the hot spots of ecosystem services and their respective socioeconomic backgrounds.

Ecological zoning is a key step in implementing differentiated policies in the process of ecosystem protection and restoration because it helps to clarify the difference in ecological functions and create suitable compensation rules for the ecological benefits [59]. According to the characteristics of different ecological zones, the government can propose corresponding strategies and suggestions for resource distribution [60]. However, the existing "National Ecological Function Zoning" (NEFZ) in China was applied to the large and important ecological sites at the national level. It is of certain significance to the protection of ecological space, while because of the lack of division in each district or county, the guidance for the future development of cities is still insufficient [61]. Our result can be regarded as a reference guiding spatial planning and ecological conservation at the county scale, and it gives insights into other highly urbanized regions.

**Supplementary Materials:** The following supporting information can be downloaded at: https://www.mdpi.com/article/10.3390/land12040733/s1. The used models and detailed parameters can be found in the Supplementary Document. References [62–68] are cited in the Supplementary Materials.

**Author Contributions:** Conceptualization, W.H. and W.Z.; methodology, W.Z. and W.H.; validation, W.Z., L.Z., Y.X., and C.L.; formal analysis, J.L.; investigation, J.L.; data curation, L.Z.; writing—original draft preparation, W.Z.; writing—review and editing, W.H.; supervision, Y.X. and W.H.; project administration, L.Z.; funding acquisition, W.H. All authors have read and agreed to the published version of the manuscript.

**Funding:** This research was funded by the Fundamental Research Funds for Central Public-Interest Scientific Institution, grant number AR2113.

**Institutional Review Board Statement:** Not applicable.

**Informed Consent Statement:** Not applicable.

**Data Availability Statement:** The data used for the research reported in this paper is available from the corresponding author upon reasonable request.

**Conflicts of Interest:** The authors declare no conflict of interest.

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
