# Peer review of "Zoning for Spatial Conservation and Restoration Based on Ecosystem Services in Highly Urbanized Region: A Case Study in Beijing-Tianjin-Hebei, China"

_land, doi:10.3390/land12040733_

Round 1
Reviewer 1 Report
I congratulate you on this very interesting research. However, I suggest recommendations to improve your work. Attached is the PDF file with the suggestions.

Author Response
We thank the reviewers for providing constructive feedback. We have fully revised our manuscript and have addressed all of the reviewers’ comments, as well as added new analyses to further strengthen our work.
The major revisions and new analyses we have undertaken are summarized below and discussed in detail in the point-by-point responses.
Point 1. I suggest starting this section with two general paragraphs: 1) related to the behavior of ecosystems in highly urbanized sectors and 2) related to zoning for spatial conservation of the ecosystem, to complement the state of the art of your research.
Response: We rewrite the introduction section including the ecological situation of urbanized areas and the necessity of ecological zoning.
Point 2. I feel like lines 58-59 state a conclusion. I suggest reviewing this idea and writing in terms of the section.
Response: We delete the sentence and add a new one in terms of the section. (Line 59-60)
Point 3. These data from the study area are very interesting. I suggest justifying them with bibliographical references.
Response: References added. (Line 113-114)
Point 4. I suggest justifying these parameters with bibliographic citations.
Response: References added. (Line 57-163)
Point 5. I suggest briefly explain the maps (a,b,c,d)
Response: Brief introductions have been added to Figure 3.
Point 6. I suggest briefly explain the maps (WR, SR, HM, CS)
Response: Brief introductions have been added to Figure 4.
Point 7. I suggest highlighting the percentage goodness-of-fit of your study's correlation analysis.
Response: We added the confidence level of Moran index. (Line 210)
Point 8. In general terms, this section talks a lot about his study, but it does not discuss his results. I suggest a discussion based on similar studies with the pros and cons of their results, supported by bibliographic citations.
Response: We have calculated six factors (precipitation, DEM, Temperature, population, proportion of built-up area, GDP) of the different zones and further discussed the potential development strategies. More references are added to support this section.
Point 9. In this section, he repeats his results. I suggest three conclusions from your study without the need to repeat the information
Response: We have summarized our research results and rewritten the conclusion section.

Reviewer 2 Report
This paper presents an interesting topic regarding zoning for spatial conservation and restoration based on ecosystem services. The language and style of writing is clear and adhere to academic standards.
The introduction of the paper is good and provide basic background data regarding the challenges of urbanization within BTH region. Unfortunately, the aim of the paper in the introductory part is not clearly presented and not correlated with conclusion of the paper.
From the row 71-82, it has to be in more detail elaborated. Why you use InVEST models? Why you use only four selected ecosystem services? Finally, how the aim of the paper reflected for guiding the strategic development and optimizing spatial zoning? The conclusion part only present statistical analysis.
The central part of the paper, which presents the statistical tests, is done correctly and represents a good basis for the analysis, however, the fourth chapter "discussion" is not based on the achieved results of the statistical analysis. Therefore, it is suggested in chapter 4 to rely on the results of the analysis and not to highlight "general facts".
I suggest the author’s work on these issues and submit a new more detailed version of the paper to the journal when ready.
Author Response
We thank the reviewers for providing constructive feedback. We have fully revised our manuscript and have addressed all of the reviewers’ comments, as well as added new analyses to further strengthen our work.
The major revisions and new analyses we have undertaken are summarized below and discussed in detail in the point-by-point responses.
Point 1. The introduction of the paper is good and provide basic background data regarding the challenges of urbanization within BTH region. Unfortunately, the aim of the paper in the introductory part is not clearly presented and not correlated with conclusion of the paper.
Response: we have modified the introduction part and the aim is rephrased to be clearly presented.
Point 2. From the row 71-82, it has to be in more detail elaborated. Why you use InVEST models? Why you use only four selected ecosystem services? Finally, how the aim of the paper reflected for guiding the strategic development and optimizing spatial zoning? The conclusion part only present statistical analysis.
Response: (1) we described the reasons for selecting four ecosystem services and InVEST models in detail (see Line120-132). (2) The conclusion has been rewritten to describe the potential application of the results.
Point 3. The central part of the paper, which presents the statistical tests, is done correctly and represents a good basis for the analysis, however, the fourth chapter "discussion" is not based on the achieved results of the statistical analysis. Therefore, it is suggested in chapter 4 to rely on the results of the analysis and not to highlight "general facts".
Response: A new table 4 was added in the discussion section. Further analysis based on the achieved results of hotspots and the statistical results in table 4 was presented to support conservation and restoration strategies of different ecological zones.
Reviewer 3 Report
The article is original and brings new knowledge to the field of ecosystem services analysis. It proposes an interesting research methodology and valuable planning guidelines for sustainable development at the regional scale.
The study's assumptions, methodology and obtained results are well described. The only thing that would require a more detailed presentation of the study area, especially in the context of the described research results (name of the sea, mountains, reservoirs, etc.). It also raises the question of whether an analysis of ecosystem services in correlation with the urbanization index was undertaken. After all, the variation of ecosystem services in space is natural and will always be different in the mountains than in the flatlands. The question remains to what extent human activity shapes them. Only this, it seems, should be the basis for formulating design guidelines for planners. In other words, is the formulation "Ecological restoration zone was composed of the counties located in the plain area where the vegetation coverage was low and the human activities were intense" (246-247) supported by the obtained research results?
As for the discussion of the results, there is basically no discussion in this part of the study (not a single reference to the literature). Only in Section 4.1 are the conservation and restoration strategies of different ecological zones formulated, and in Section 4.2 "Limitations and future perspectives." Thus, there is essentially no discussion.
Conclusions are drawn from the analyses and synthesize what the article is about.
Author Response
We thank the reviewers for providing constructive feedback. We have fully revised our manuscript and have addressed all of the reviewers’ comments, as well as added new analyses to further strengthen our work.
The major revisions and new analyses we have undertaken are summarized below and discussed in detail in the point-by-point responses.
Point 1. The study's assumptions, methodology and obtained results are well described. The only thing that would require a more detailed presentation of the study area, especially in the context of the described research results (name of the sea, mountains, reservoirs, etc.). It also raises the question of whether an analysis of ecosystem services in correlation with the urbanization index was undertaken. After all, the variation of ecosystem services in space is natural and will always be different in the mountains than in the flatlands. The question remains to what extent human activity shapes them. Only this, it seems, should be the basis for formulating design guidelines for planners. In other words, is the formulation "Ecological restoration zone was composed of the counties located in the plain area where the vegetation coverage was low and the human activities were intense" (246-247) supported by the obtained research results?
Response: We added the name of sea, mountains, reservoirs in the Study area introduction (Line 89-92). In the discussion part we added a new table 4 calculating the natural and artificial factors in the ecological zones to show the correlation between ecosystem services and urbanization level. Additionally, we discussed how these factors could be joint analyzed to support conservation and restoration strategies of different ecological zones (Section 4). We have also deleted the sentence "Ecological restoration zone was composed of the counties located in the plain area where the vegetation coverage was low and the human activities were intense". Detailed description about ecological restoration zone was added in the discussion section (Line 284-287).
Point 2. As for the discussion of the results, there is basically no discussion in this part of the study (not a single reference to the literature). Only in Section 4.1 are the conservation and restoration strategies of different ecological zones formulated, and in Section 4.2 "Limitations and future perspectives." Thus, there is essentially no discussion.
Response: We have enriched the discussion section by adding a new table 4 and more references to do a joint analysis to support development strategies of different ecological zones.
Point 3. Conclusions are drawn from the analyses and synthesize what the article is about.
Response: We have rewritten the conclusion section.
Reviewer 4 Report
Dear authors of the manusript,
thank you very much for the submission to land. It would benefit from intensive grammar and language check - the abstract itself is difficult to understand. Looking at the context, the work presents yet another INVEST modelling (a trend for many years in china). Novelty is very low, as the work presents a case study - I would suggest finding a local journal to publish this - as there have been many many many studies of this kind, with the similar structure, similar invest formular. All the best with the manuscript.
Author Response
We thank the reviewers for providing constructive feedback. We have fully revised our manuscript and have addressed all of the reviewers’ comments, as well as added new analyses to further strengthen our work.
Comment: thank you very much for the submission to land. It would benefit from intensive grammar and language check - the abstract itself is difficult to understand. Looking at the context, the work presents yet another INVEST modelling (a trend for many years in china). Novelty is very low, as the work presents a case study - I would suggest finding a local journal to publish this - as there have been many many many studies of this kind, with the similar structure, similar invest formular. All the best with the manuscript.
Response: the abstract has been rewritten and the whole text was proofread and checked. In this paper, we adopted the newly produced high-resolution landcover data and such zoning analysis by using INVEST models for BTH region is seldom. We believe this zoning approach could be interested for the local planners or practitioners.
Reviewer 5 Report
Reviewers comments
The study of Zhou et al., deals with conservation and restoration of Ecosystems services (ES) in urbanized area. Topic is of great importance and relevant, as maintaining in urbanized areas is essential for resilience of cities and human well-being. However, I made some suggestions and detect some minors mistake authors must consider before their work being accepted and published in this journal:
1) Authors must explain clearly the term " Coldspot" used in Material and Method;
2) They made an analysis for "Hotspot", but nothing is clearly said on "Coldspot" and it relevance in the present study;
3) I Suggest you includes data on population density or population size of the all the BTH counties
4)Line 87: Why area size is presented like this " 21.69×104 km 2 " and not like this " 2255.76 km 2 ?
5)Line 141: Replace "the" by " a "
6) Line 145 & 147: These group of word are not sentences, verb are missing so you need not a ful stop at the end. Please, rewrite this part.
7) Line 165-166: lease explain a bit why you multiply each value calculated by 108, 102tg
8)Line: 276: Have you data to support this assertion? Nothing is said on population density in your Material and Methods section
9) Line 287-288: Need a reference here for this assertion. The whole discussion is also without cited reference. Are you conduct the work for the first time in China ? Is writing the whole discussion without the cited reference a requirement of Journal?
10) Lines 327-329: We need not values here, but the ecological significance of all these number must be reported in conclusion. Just report the important significance of your results not repeat the results again

Author Response
We thank the reviewers for providing constructive feedback. We have fully revised our manuscript and have addressed all of the reviewers’ comments, as well as added new analyses to further strengthen our work.
The major revisions and new analyses we have undertaken are summarized below and discussed in detail in the point-by-point responses.
Point 1. Authors must explain clearly the term " Coldspot" used in Material and Method;
Response: We explained the concept of coldspot. (Line 149-150).
Point 2. They made an analysis for "Hotspot", but nothing is clearly said on "Coldspot" and it relevance in the present study;
Response: The coldspots are shown in Figure 5 and statistical results are explained in Table 2. However, only the hotspots of ecosystem services were used in spatial zoning, since the coldspots were not significant related to function-oriented zoning. We have clearly explained this in Line 217-218.
Point 3. I Suggest you includes data on population density or population size of the all the BTH counties
Response: A new table 4 was added in discussion including population density.
Point 4. Line 87: Why area size is presented like this " 21.69×104 km 2 " and not like this " 2255.76 km 2 ?
Response: We changed the number expression.
Point 5. : Replace "the" by " a "
Response: We changed the word.
Point 6. Line 145 & 147: These group of word are not sentences, verb are missing so you need not a ful stop at the end. Please, rewrite this part.
Response: We have rephrased this part. (Line 157-162)
Point 7. Line 165-166: Please explain a bit why you multiply each value calculated by 108, 102tg
Response: Multiplied by 108, 102tg is a writing error. We have changed to 97.4×108m3 (WR), 5.53×108t (SR), 2,673 tg (CS).
Point 8. Line: 276: Have you data to support this assertion? Nothing is said on population density in your Material and Methods section
Response: We added the statistic of six factors (precipitation, DEM, Temperature, population, Proportion of construction land area, GDP) to support the discussion, see table 4.
Point 9. Line 287-288: Need a reference here for this assertion. The whole discussion is also without cited reference. Are you conduct the work for the first time in China ? Is writing the whole discussion without the cited reference a requirement of Journal?
Response: We have rewritten the discussion section and added more references.
Point 10. Lines 327-329: We need not values here, but the ecological significance of all these number must be reported in conclusion. Just report the important significance of your results not repeat the results again.
Response: The conclusion has been rewritten.
Round 2
Reviewer 4 Report
Dear Authors,
thanks again for the resubmission of the manuscript. I think the work is well written and improved, yet, I do not understand how the core of the manuscript works. How is the zonig created as there is no methodological explanation except hot and coldspot analysis. I find the basis of the manuscript weak, basing zoning on modelling of INVEST (needs to be in the abstract!), without providing the input data for the modelling itself (absolute no go! how shall one see whether you applied the default data for the US and not regionally specific data? it cannot be accept this without it.) Plus, Table 1 speaks of values - what are values - is it modelling results? And how is the zoning done - why these areas?
I simply dont see the need for a zoning based on water retention, soil retention, heat mitigation and carbon storage. Who will find this information useful? Why these ES? It all seems a bit quick and arbitrary to me.
I suggest to submit the manuscript to a more regional journal.
Author Response
Point 1.How the core of the manuscript works. How is the zonig created as there is no methodological explanation except hot and coldspot analysis.
Response: We used the K-means clustering on the hotspots of ecosystem services for spatial zonation at the county level in the BTH region. The K-means algorithm is supplied by Geoda software. The number of clusters was decided by considering of average variances of ecosystem services in each zone as a minimum. Please find the detail in section 2.4.
Point 2.I find the basis of the manuscript weak, basing zoning on modelling of INVEST (needs to be in the abstract!), without providing the input data for the modelling itself (absolute no go! how shall one see whether you applied the default data for the US and not regionally specific data? it cannot be accept this without it.)
Response: we adopt the latest and most detailed land cover data from the national project of Geographical Conditions Monitoring (GCM) which is more suitable for spatially modelling ecosystem services (Line 74-76). At the same time, we used the regionally specific data as input data to calculate parameters in each ecosystem service function, including rainfall data, soil data, meteorological data, etc. The specific data sources and parameters can be found in the Supplementary material..
Point 3.Plus, Table 1 speaks of values - what are values - is it modelling results? And how is the zoning done - why these areas?
Response: Table 1 shows the statistical value of the modelling results of different ecosystem services in the BTH region. We have changed the caption of Table 1. Four administrative units (Beijing city, Tianjin city, Hebei Province, and the whole BTH region) was used to quantitatively compare the differences of their ecosystem services.
Point 4.I simply dont see the need for a zoning based on water retention, soil retention, heat mitigation and carbon storage. Who will find this information useful? Why these ES?
Response: The ecosystem services are selected based on the urgent demands for the ecosystem services of BTH region (see Line 123-135), as well as the available data. On the other hand, we focus on the regulation and maintenance services as they are mostly considered for compensation for ecological protection, since provisioning and cultural services are driven by human needs and produce economic gains.
Compared to the “National Ecological Function Zoning” (NFFZ) published in 2008, our results are much detailed and can be regarded as a reference guiding the spatial planning and ecological conservation at the county scale, as well as gives insights for other highly urbanized regions. (Line 372-383).
Reviewer 5 Report
I have no more comments
All the suggestions have been integrated in the text by the Authors
Author Response
Thank you very much for your review again!